# Mechanosensitive localization of Diversin highlights its function in vertebrate morphogenesis and planar cell polarity

Satheeja Santhi Velayudhan, Chih-Wen Chu, Keiji Itoh and Sergei Y. Sokol*

## ABSTRACT

Diversin is a vertebrate homolog of the core planar cell polarity (PCP) protein Diego. Here we studied the function of Diversin in *Xenopus* embryo morphogenesis and its subcellular localization at different locations in superficial ectoderm cells. Depletion of Diversin in the neuroectoderm inhibited apical domain size and neural tube closure and disrupted the polarized localization of endogenous Vangl2, another PCP protein. Whereas Diversin puncta were randomly distributed in early ectoderm, they acquired planar polarity in the neuroectoderm in a stage- and position-specific manner. We find that Diversin is accumulated at the cell junctions adjacent to apically constricting cells at the *Xenopus* neural plate border and the gastrula blastopore lip. Moreover, Diversin cytoplasmic puncta redistributed in the direction of the pulling forces from the cells with constricting apical domains, suggesting a mechanosensitive process. PCP complexes of Dishevelled (Dvl2) and Diversin or the mechanosensitive adaptor ADIP exhibited planar polarity in the neural plate and the wound edge and promoted wound healing. We propose that Diversin- and Dvl2-containing PCP complexes control morphogenesis in a tension-dependent manner.

KEY WORDS: Diversin, Ankrd6, Planar cell polarity, Mechanosensation, Neural tube defects, Dishevelled, Dvl2

## INTRODUCTION

Planar cell polarity (PCP) is a common phenomenon that coordinates cell orientation within tissue plane. PCP is manifested by the asymmetric distribution of conserved core PCP proteins in epithelial tissues (Butler and Wallingford, 2017; Gray et al., 2011; Sokol, 2015). In *Drosophila*, these include the transmembrane proteins Frizzled (Fz), Strabismus/Vangl, and Flamingo/Celsr, and the cytoplasmic proteins Dishevelled (Dvl), Diego, and Prickle. In vertebrates, core PCP components also asymmetrically localize within the neuroectoderm (Chu and Sokol, 2016; Ciruna et al., 2006; Ossipova et al., 2015b; Yin et al., 2008). PCP signaling is believed to be essential for morphogenetic processes in vertebrate embryos, including gastrulation and neurulation (Butler and Wallingford, 2017; Goodrich and Strutt, 2011; Gray et al., 2011),

Department of Cell, Developmental and Regenerative Biology, Icahn School of Medicine at Mount Sinai, New York, NY 10029, USA.

*Author for correspondence (sergei.sokol@mssm.edu)

S.Y.S., 0000-0002-3963-9202

however, the underlying downstream molecular mechanisms leading to morphogenesis remain poorly understood.

Diversin (also known as Ankrd6) is a vertebrate homolog of the fly planar cell polarity protein Diego (Moeller et al., 2006; Schwarz-Romond et al., 2002). In *Drosophila*, Diego interacts with Dvl, Vang and Prickle to compete with Prickle for Dvl binding and stimulate polarized cell behaviors via Fz/Dvl activity in the wing epithelium (Das et al., 2004; Jenny et al., 2005). In vertebrates, Diversin was similarly implicated in PCP and was proposed to inhibit the canonical Wnt pathway by promoting β-catenin degradation (Itoh et al., 2009; Schwarz-Romond et al., 2002). Consistent with its function in PCP, Diversin polarizes in mouse inner ear sensory cells (Jones et al., 2014) and frog neuroectoderm (Ossipova et al., 2014) and regulates axis elongation in zebrafish embryos via its interaction with Dvl (Moeller et al., 2006; Schwarz-Romond et al., 2002). Our previous studies have shown that Diversin associates with the centrosome and the basal body (Itoh et al., 2009; Yasunaga et al., 2011). Diversin depletion revealed its requirement for ciliogenesis in multiciliated skin cells and gastrocoel roof plate cells, and its role in left–right asymmetry (Yasunaga et al., 2011). Another Diego homolog, mouse Inversin, similarly regulates left–right asymmetry and kidney development (Mochizuki et al., 1998), acting as a switch between Wnt signaling branches (Simons et al., 2005).

Diversin is abundant in mammalian (mouse) CNS (Tissir et al., 2002) and, like other core PCP proteins, has been linked to the incidence of human neural tube defects (Allache et al., 2015) and neural tube closure in *Xenopus* embryos (Ossipova et al., 2014). We recently discovered the association of Diversin with the microtubule-binding protein ADIP (Afadin- and alpha-actinin-binding protein, also known as SSX2IP and Msd1) during wound healing and the planar polarization of the Diversin-ADIP complex in the plane of epithelia (Chu et al., 2025). Since ADIP is a mechanosensitive protein (Chu et al., 2025), we hypothesize that Diversin is also involved in the response of embryonic epithelia to physical forces and examined its localization and function in the *Xenopus* neuroectoderm.

Our loss-of-function analysis suggests that Diversin is required for apical constriction of neural plate cells. Neuroepithelial cells depleted of Diversin exhibit abnormal polarization of Vangl2, consistent with the proposed function of Diversin in PCP. Polarized cortical enrichment of Diversin in surface epithelial cells at the locations subjected to pulling forces suggests that Diversin can respond to mechanical tension. This hypothesis was confirmed by redistribution of Diversin in response to forces produced by apically constricting cells. We also demonstrate that Dvl2, another core PCP protein, forms complexes with Diversin and ADIP that polarize in normal morphogenetic events and during wound healing. Our findings lead us to propose that Diversin and Dvl2 are involved in the mechanosensitive segregation of PCP protein complexes in the neural plate.

## RESULTS

### Diversin is required for apical constriction and Vangl2 planar polarization in *Xenopus* neuroectoderm

To investigate a role of Diversin in morphogenetic events, we performed unilateral knockdown of Diversin by injecting a previously characterized antisense morpholino (DivMO) (Yasunaga et al., 2011) in dorsal blastomeres of four-cell *Xenopus* embryos (Fig. 1A).

At stage 16, unilateral Diversin knockdown exhibited wider neural plate area at the injected side (Fig. 1B,C) that resulted in neural tube closure defects (Fig. 1D-E′), confirming previous observations (Ossipova et al., 2014). At stage 16, DivMO-injected cells exhibited a marked reduction in F-actin accumulation compared to the uninjected or control morpholino (CoMO)-injected controls (Fig. 1B′), consistent with the link between reduced cortical and junctional actin and impaired epithelial cell

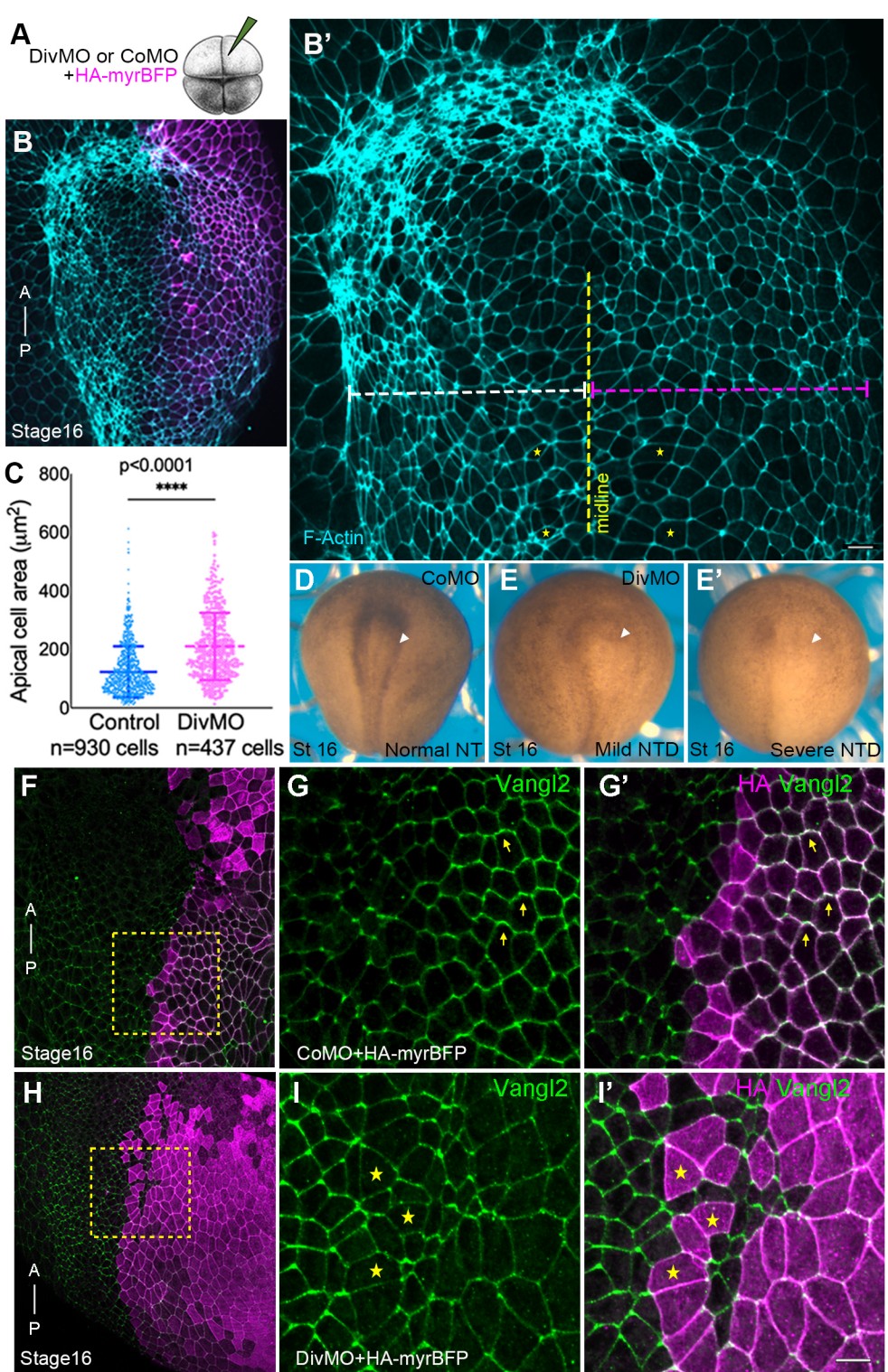

**Fig. 1. Diversin depletion impairs apical constriction and leads to planar polarity defects in the neuroectoderm.**
(A) Experimental scheme. Diversin (Div) morpholino (MO) or control MO (CoMO) was injected with HA-myr-BFP RNA as lineage tracer into a single dorsal blastomere of four-cell *Xenopus* embryos.
(B,C) Diversin depletion inhibits neural plate folding, reduces F-actin and expands the apical domain.
(B) Representative stage 16 embryo shows unilaterally expanded neural plate and reduced F-actin intensity in DivMO-injected cells marked by HA-myr-BFP (magenta). (B′) Magnified view of phalloidin-stained cells illustrating the expanded apical domain in DivMO-injected cells (asterisks) on the right side, as indicated by the dashed magenta line.
(C) Quantification of the apical domain area; total numbers of cells per group are indicated. ****$P<0.0001$, Student's $t$-test.
(D-E′) Representative phenotypes of stage 16 embryos unilaterally injected at the four-cell stage with 30 ng of either control morpholino (CoMO) or Diversin MO (DivMO). Arrowheads indicate the injected side. Compared to the normal neural tube closure in CoMO-injected embryos (D), DivMO-injected embryos exhibited a wide range of defects, from mild abnormalities with weak or discontinuous pigmentation (E) to severe neural tube folding defects (E′).
(F-I′) Defective Vangl2 polarity in Diversin-depleted embryos. Cells of stage 16 embryo injected with DivMO are marked with HA-myr-BFP (magenta). Anteroposterior (A–P) axis is indicated.
(F,H) Vangl2 immunostaining in the magnified regions of CoMO- (F) and DivMO-injected (H) embryos (dashed boxes). (G-G′) Vangl2 is polarized in CoMO-injected cells. (I-I′) Vangl2 polarity is reduced in DivMO-injected cells (asterisks). Anti-HA staining for HA-myr-BFP (F,G′,H,I′), and anti-Vangl2 staining (G,I) are shown. HA-myr-BFP staining is variable for F and H, due to distinct locations of the injection sites. Scale bars: 20 μm. Data represent six embryos from three independent experiments.

shape during neurulation (Haigo et al., 2003; Martin and Goldstein, 2014). Supporting this hypothesis, Diversin-depleted cells display enlarged apical domains, indicative of defective apical constriction (Fig. 1B,C). Immunostaining for the core PCP component Vangl2 revealed normal anterior–posterior planar polarization in CoMO-injected controls (Fig. 1F-G′), whereas Vangl2 did not polarize in Diversin-depleted cells (Fig. 1H-I′).

These observations establish an essential role of Diversin for Vangl2 planar polarization and reveal its function in actomyosin contractility in the neuroectoderm.

### Diversin marks local PCP in neural and non-neural ectoderm and in the superficial cells adjacent to the blastopore

We next assessed Diversin protein localization in the ectoderm of *Xenopus* embryos expressing GFP-Diversin RNA (Fig. 2A). At stage 10.5 and 12, Diversin was detected as uniformly distributed puncta at the membrane, cortex, and cytoplasm of ectoderm cells (Fig. 2B-D). By stage 14, the protein puncta accumulated towards the neural plate midline (Fig. 2E-H). At stage 16, Diversin puncta became enriched at the junctions of epidermal cells that were proximal to the neural plate border (Fig. 2I-L). Planar polarity of Diversin puncta extended through several rows of cells, suggestive of signaling between the epithelial cells.

We also examined the subcellular distribution of Diversin at stage 10.5, when gastrulation movements are actively underway. At this stage, the blastopore forms in the marginal zone of the embryo (Fig. 3A). RFP-Diversin displayed random localization in animal cap ectoderm cells (Fig. 3B) but became clearly polarized at the cell corners proximal to the blastopore lip, both in the dorsal and ventral

marginal zone (Fig. 3C,D). Since Diversin interacts with ADIP, a force-responsive adaptor protein (Chu et al., 2025), the observed dynamic developmentally regulated segregation of Diversin in the neural plate and near the blastopore suggests that, like ADIP, Diversin responds to mechanical cues during gastrulation or neurulation.

### Mechanosensitivity of Diversin protein localization in response to Shroom3 or Plekhg5

To support the hypothesis that *in vivo* Diversin localization reflects its mechanosensing properties, we induced apical constriction in *Xenopus* ectodermal cell clones by local injection of mRNAs encoding apical constriction inducers, Shroom3 (Haigo et al., 2003; Hildebrand and Soriano, 1999) or Plekhg5 (Popov et al., 2018) (Fig. 4A). Diversin localization was assessed in the tissue surrounding the apically constricting cells. Whereas Diversin puncta distributed randomly in the control ectoderm (Fig. 4B,C), they became enriched along the cell borders adjacent to the clones of Shroom3- or Plekhg5-expressing cells (Fig. 4D-G).

These findings suggest that Diversin puncta can sense mechanical cues from neighboring tissues and become planar polarized in response to such cues.

### Diversin colocalizes and polarizes with Dvl2 in the neural plate

Given the observed planar polarity of Diversin puncta, we wanted to know the localization of other core PCP proteins. Dvl has a characteristic segregation in *Drosophila* epithelia to one side of the cells (Axelrod, 2001). A similar bias in the distribution of vertebrate

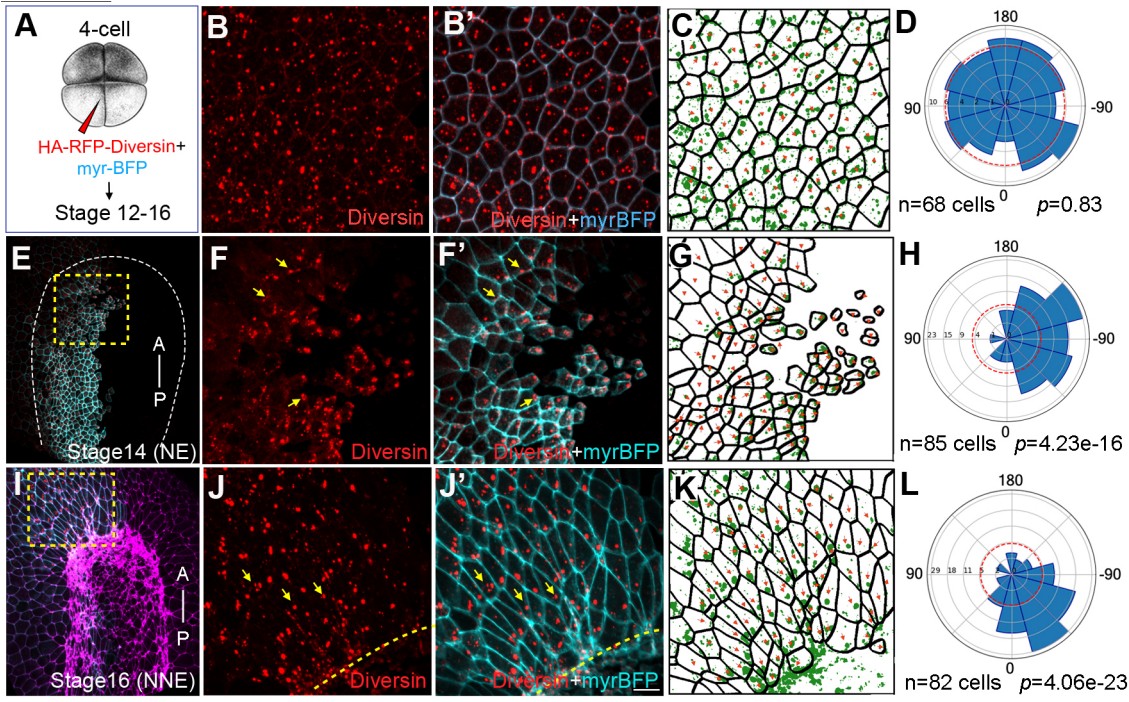

**Fig. 2. Planar polarization of Diversin in ectoderm cells during neurulation.** (A) Schematic of RNA injection: HA-RFP-Diversin RNA (200 pg) was co-injected with myr-BFP RNA (50 pg) as cell border and lineage tracer into four-cell embryos to target neural or non-neural ectoderm. (B-D) Animal pole ectoderm at stage 12. (E-H) Anterior neural plate at stages 14. (I-L) Nonneural ectoderm adjacent to anterior neural plate border. Yellow arrows indicate Diversin accumulation at cell corners. Dashed boxes highlight regions with polarized Diversin that are magnified in F and J. Neural plate border is marked by dashed lines or intense F-actin staining (magenta in I). Anterior–posterior axis is shown in (E,I). NE and NNE, neural and non-neural ectoderm, respectively. Scale bar: 20 μm. (C,G,K) Cell segmentation. Red arrows indicate cell polarity marked by Diversin puncta enrichment. Rose plots (D,H,L) show Diversin puncta enrichment relative to the posterior axis (0° in H), neural plate midline (H), or anterior neural plate border (L). Data are from three embryos. Chi-square tests indicate non-random Diversin distribution.

Biology Open

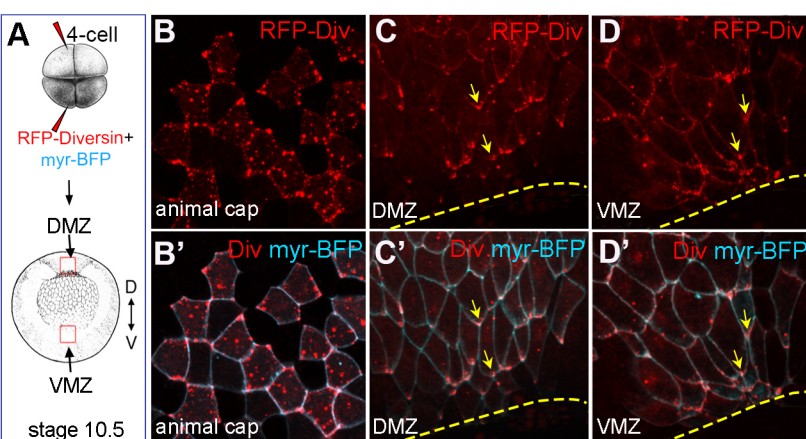

**Fig. 3. Diversin polarizes in the cells adjacent to the blastopore lip during gastrulation.** (A) Schematic of RNA injections. RFP-Diversin (200 pg) and myr-BFP (50 pg) mRNAs were co-injected into the animal pole or the marginal zone of *Xenopus* four-cell embryos. (B) Animal view of early gastrula, (C-D′) vegetal view of stage 10.5 gastrula with red dashed boxes corresponding to the dorsal and ventral marginal zones (DMZ and VMZ) at the blastopore lip. (B-D″) RFP-Diversin localization in animal ectodermal cells (B,B′), or cells adjacent to the DMZ (C,C′) and VMZ (D,D′) blastopore lip at stage 10.5. Yellow arrows indicate asymmetric Diversin accumulation at cell corners. Scale bar: 50 μm. Data represent five to ten embryos per group.

Dvl homologues was reported for mammalian cells and zebrafish embryos (Vladar et al., 2012; Yin et al., 2008). Since Dvl associates with Diversin in zebrafish embryos (Moeller et al., 2006), we expressed RNAs encoding RFP-Diversin, GFP-Dvl2, and myr-BFP and assessed protein distribution in the area of the neural plate (Fig. 5A). When injected on its own, GFP-Dvl2 puncta exhibited weak accumulation in the neural plate cells toward the dorsal midline (Fig. 5B,B′). In stage 12 ectoderm cells, GFP-Dvl2 fully colocalized with RFP-Diversin (Fig. 5C-C″), consistent with their interaction (Moeller et al., 2006). At stage 14, the complexes of Dvl2 and Diversin became asymmetrically enriched in anterior neural ectodermal cells, orienting towards the dorsal midline (Fig. 5E-F′). In non-neural ectodermal cells, Diversin and Dvl2 puncta became oriented towards the anterior neural plate border

(Fig. 5G-H′). This polarization suggests that the two proteins function together to establish PCP.

## Tension-responsive complex of Dvl2 and ADIP in wound healing

Since our observations implicated Dvl2 in tension-dependent PCP, we wanted to evaluate whether this protein is involved in other force-dependent processes. We have recently shown that Diversin interacts with the microtubule-binding protein ADIP and the ADIP-Diversin complex polarizes during wound healing (Chu et al., 2025). We therefore decided to assess a role of the ADIP-Dvl2 complex in wound healing. In stage 11 ectoderm, the cytoplasmic puncta of Dvl2 colocalized with GFP-ADIP when the two proteins were co-expressed (Fig. 6A-A″). Upon wound healing,

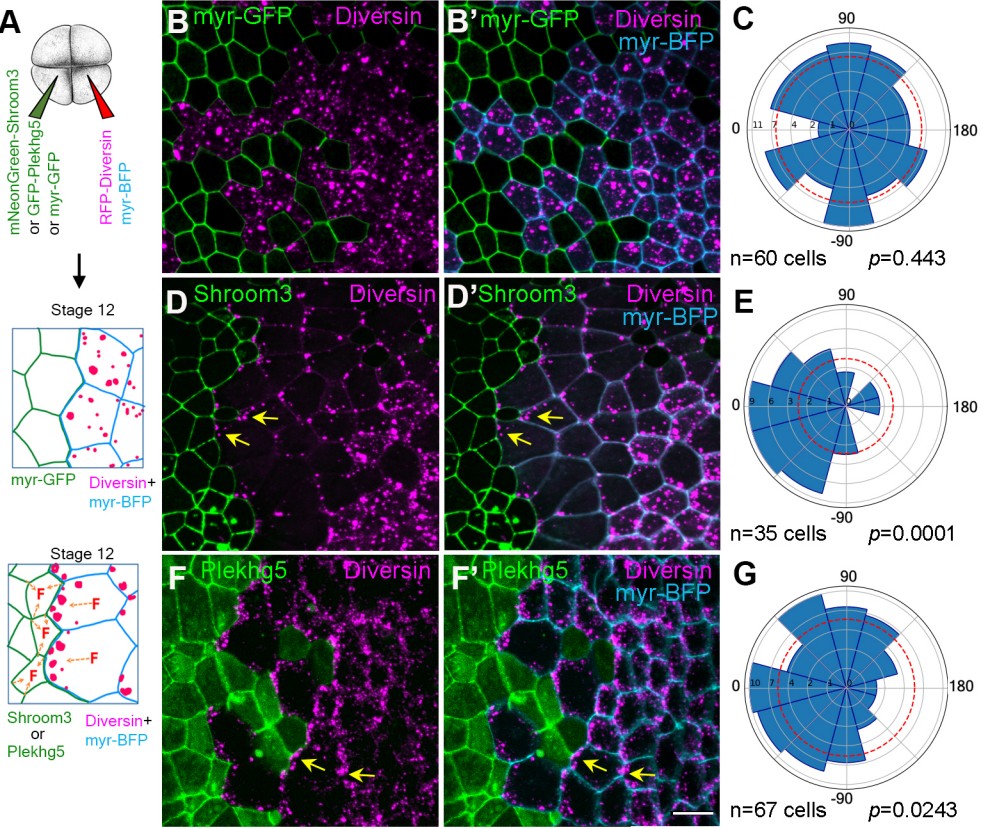

**Fig. 4. Diversin puncta orient towards apically constricting neighboring cells.** (A) Experimental scheme. Clones expressing control myr-GFP (20 pg), mNeonGreen-Shroom3 (20 pg), or GFP-Plekhg5 (20 pg) mRNA. RFP-Diversin-expressing cells are marked with myr-BFP (cyan). (B-B″,D-D″,F,F′) Random distribution of RFP-Diversin (red) in cells adjacent to control mem-GFP clones at stage 12.5. (B,B′) Polarized accumulation of RFP-Diversin toward neighboring mNeonGreen-Shroom3 (D,D′), and GFP-Plekhg5 (F,F′) expressing cells. Yellow arrows show the enrichment of Diversin puncta. Scale bar: 20 μm. (C,E,G) Rose plots show Diversin puncta orientation relative to the border of adjacent control cells (C), mNeonGreen-Shroom3-expressing cells (E) and GFP-Plekhg5-expressing cells (G), respectively. (0° is perpendicular to the border). Data represent three embryos per condition. Chi–square analysis confirms random distribution in controls (B-B″,C) but not in the cells adjacent to Shroom3 (D-D″,E) or Plekhg5-expressing cells (F-F″,G), respectively.

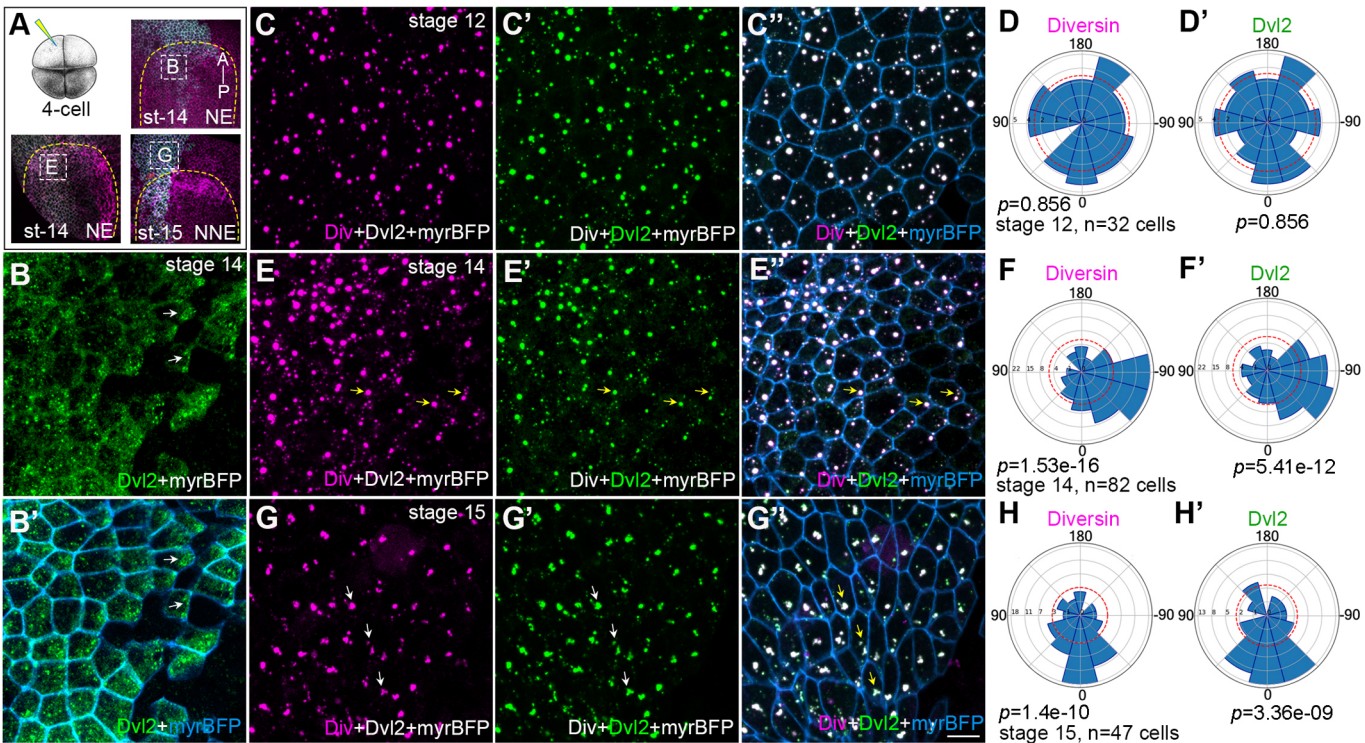

**Fig. 5. Diversin colocalizes with Dvl2 in neural and epidermal ectoderm.** (A) Injection scheme. GFP-Dvl2 (200 pg) together with myr-BFP (50 pg) or RFP-Diversin (200 pg), GFP-Dvl2 (200 pg) with myr-BFP (50 pg) mRNAs were co-injected as indicated into one dorsal blastomere of four-cell stage *Xenopus* embryos. Myr-BFP marks cell borders. Neural plate border is outlined by yellow dashed lines, Phalloidin staining is shown (magenta), and the anterior–posterior (A–P) axis is indicated. White boxes mark stage 14 anterior neural ectoderm (NE) in (B-B′) and (E-E″) or adjacent stage 15 non-neural ectoderm (NNE) in (G-G‴). (B,B′) Stage 14 anterior neural ectoderm, dorsal view. GFP-Dvl2 (B); merged with myr-BFP (B′). GFP-Dvl2 puncta weakly oriented in few cells relative to midline of the neural plate. (C-D′) Stage 12 ectoderm. RFP-Diversin (C); GFP-Dvl2 (C′); merged with myr-BFP (C″). Rose plots show random orientation of Diversin (D) and Dvl2 (D′). (E-F′) Stage 14 anterior neural ectoderm embryo show the localization of RFP-Diversin (E), GFP-Dvl2 (E′); merged with myr-BFP (E″). Rose plots show orientation of Diversin (F) and Dvl2 (F′) to the midline of neural plate. (G-H′) Stage 15 anterior non-neural ectoderm, dorsal view. RFP-Diversin (G), GFP-Dvl2 (G′); merged with myr-BFP in G″. White arrows show the orientation of Diversin and Dvl2. Scale bar: 20 µm. (E-G′), Rose plots show Diversin (H) and Dvl2 puncta (H′) polarization in tissue plane, 0° corresponds to the posterior axis. Quantification is based on three independent embryos. Chi-squire test shows non-random distribution of Diversin and Dvl2 puncta.

the Dvl2-ADIP complex sharply oriented toward the wound edge (Fig. 6B-B″), mirroring a similar localization of the Diversin-ADIP complex (Chu et al., 2025). Immunoprecipitation analysis confirmed that Dvl2 physically binds to ADIP (Fig. 6C).

To analyze a role of Dvl2 in wound healing, we expressed a dominant-interfering Dvl2 construct (DEP+) that blocks Dvl function during collective cell movements (Tada and Smith, 2000). In the embryos expressing Dvl2-DEP+ wound healing was delayed, demonstrating a functional requirement for Dvl2 in repair (Fig. 6D,E). Together, these experiments show that both Diversin and Dvl2 form mechanosensitive complexes that direct epithelial wound healing.

## DISCUSSION
Our study identifies a unique and dynamic pattern of Diversin subcellular distribution and polarization during gastrulation and neurulation. In gastrula ectoderm, Diversin puncta are distributed randomly. Diversin becomes posteriorly oriented in the cells adjacent to the blastopore, consistent with the establishment of PCP during gastrulation (Chien et al., 2015; Mancini et al., 2021). In anterior neural plate cells, Diversin puncta are enriched towards the midline at stage 14. By stage 16, Diversin becomes enriched in non-neural ectoderm toward the neural plate border. This pattern is reminiscent of the pattern described for Rab11 (Ossipova et al., 2014)

and ADIP (Chu et al., 2025), but distinct from the segregation of other core PCP proteins, such as Vangl2 or Prickle along the anterior-posterior axis (Chu and Sokol, 2016; Ciruna et al., 2006; Devenport and Fuchs, 2008; Ossipova et al., 2015b; Yin et al., 2008). The time- and location-dependent distribution of Diversin is consistent with its potential roles in PCP signaling and in the control of early morphogenetic movements during vertebrate development. Supporting a conserved role for Diversin/Ankrd6 in PCP, mice lacking the Ankrd6 gene exhibited PCP defects in the hair cells of the inner ear (Jones et al., 2014).

The Diversin localization pattern suggests that it can sense mechanical forces acting in the early embryo during gastrulation and neurulation. Several arguments indicate that Diversin distribution is determined by tension-sensitive signaling mechanisms. First, Diversin is polarized in the embryonic tissues that undergo morphogenesis and are likely to respond to mechanical stimuli. Second, Diversin puncta accumulate at the cell junctions that are adjacent to regions of elevated tension triggered by locally induced apical constriction. Third, Diversin is polarized in the epithelium during wound healing and in response to direct tissue stretching *in vivo* (Chu et al., 2025). Together with the recent study demonstrating the polarization of ADIP-Diversin complexes during wound healing (Chu et al., 2025), our findings support the idea that PCP signaling is an integral component of mechanochemical

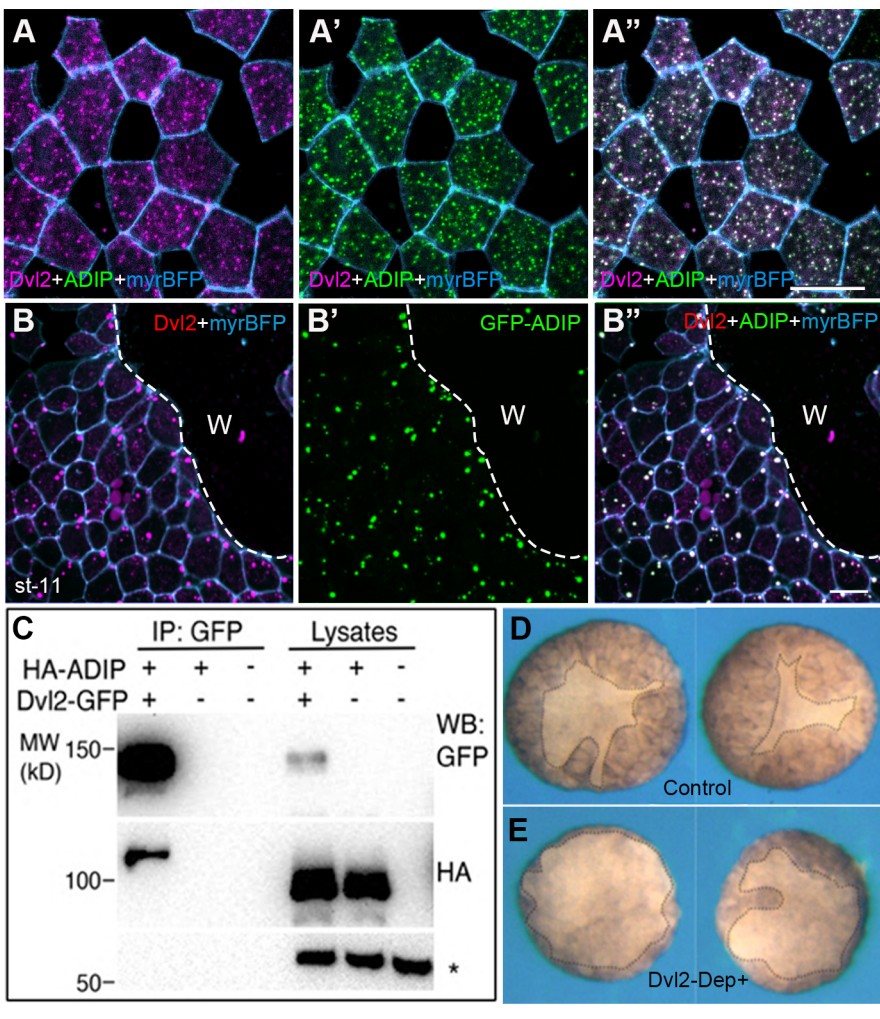

**Fig. 6. Polarization of the Dvl and ADIP complex during epidermal wound healing.**
(A-A″) Colocalization of RFP-Dvl2 (A) and GFP-ADIP (A′) in the control unwounded stage 11 *Xenopus* ectoderm. (B-B″) Dvl2 (B) and ADIP (B′) puncta polarize relative to the wound edge. Scale bars: 20 µm. Cell borders are outlined by myrBFP. Data are representative of two independent experiments. (C) Physical association of Dvl2-GFP and HARFP-ADIP revealed by the pulldown with GFP-trap. Non-specific band (asterisk) demonstrates equal protein loading. Molecular weight markers (kD) are indicated on the left. (D,E) Inhibition of epidermal wound healing by the dominant-interfering Dvl2 construct. Ectoderm explants were isolated from stage 9.5 embryos injected earlier with Dvl-DEP+ RNA (1 ng, E) or control embryos (D). Wound healing was assessed after 1 h post wounding as the area of the epidermis covering the wound (marked by the dotted line). Data are representative of three experiments with 58 explants in each group.

feedback regulation driving collective cell behaviors during morphogenesis.

Diversin properties are consistent with its role in sensing intercellular tension and converting it into planar polarity cues. Originally characterized as a Wnt pathway scaffold protein (Schwarz-Romond et al., 2002), Diversin likely acts via the proteins involved in both Wnt and planar polarity signaling, such as Dvl2. Similar to other core PCP proteins (Matsuda et al., 2023; Ossipova et al., 2015a), Diversin is cell-autonomously required for F-actin enrichment and apical constriction, and is necessary to establish PCP in the neuroectoderm, consistent with the neural tube closure and wound healing defects observed upon Diversin knockdown (Chu et al., 2025; Ossipova et al., 2014). Together with the recent reports, in which PCP correlates with the anteroposterior differences in junctional tension in frog mesoderm (Weng et al., 2025) and myosin light chain phosphorylation in the chick inner ear (Prakash et al., 2025), our work further strengthens the link between PCP and tensile forces in early development.

This study identified Dvl2 as a component of the PCP complex that responds to mechanical tension *in vivo*. Both Diversin and Dvl are established mediators of Wnt and PCP signaling pathways. In *Drosophila* wing epithelia, the interaction of Diego, a Diversin homologue, with Dsh facilitates the assembly and the segregation of core PCP complexes involving Fz and Strabismus/Vangl2 (Feiguin et al., 2001; Jenny et al., 2005). In vertebrates, Diversin was suggested to act by affecting Dvl function (Moeller et al., 2006;

Schwarz-Romond et al., 2002). Through its ankyrin repeat domains, Diversin binds Dvl (Moeller et al., 2006; Schwarz-Romond et al., 2002). Notably, our observations indicate that Diversin is a limiting factor for Dvl2 planar polarization, because Diversin colocalizes and polarizes with Dvl2 in the anterior neural plate, whereas Dvl2 alone only exhibited weak planar polarity on its own. Another factor that promotes Dvl2 planar polarity is ADIP, an adaptor binding to both actin and microtubules (Asada et al., 2003; Hori et al., 2014; Toya et al., 2007). The observed interaction and polarization of Dvl2 and ADIP in the proximity of a wound raises the possibility that Diversin and Dvl2 function with ADIP in the context of large signaling complexes that control morphogenesis. Together, our experiments suggest that anisotropic tensions generated during morphogenesis cause the recruitment of PCP protein complexes to specific junctions, locally modulate actomyosin contractility and lead to further morphogenetic changes in the neighboring cells and tissues.

## MATERIALS AND METHODS
### Plasmid constructs, RNAs and morpholinos
The plasmids pCS105-HA-RFP-Diversin, pCS2-Dvl2-GFP and pSP64T-Dvl2-HA-RFP (Itoh et al., 2005, 2009; Miller et al., 1999; Schwarz-Romond et al., 2002), pXT7-EGFP-ADIP (Reis et al., 2021), pCS2-myr-tagBFP-HA, and pCS2-myrGFP (Matsuda et al., 2023; Ossipova et al., 2014) were previously described. pCS2-myr-tagBFP-HA was generated by combining the membrane-targeting signal peptide from the Lyn kinase (MGCIKSKRKDNLNDDE), mTagBFP2 and three HA tags

(YPYDVPDYA) in pCS2 (Matsuda et al., 2023). The constructs pCS2-mNeonGreen-mShroom3 and pCS107-GFP-Diversin were generated by subcloning the cDNAs encoding mShroom3 and Diversin into the respective vectors, pCS2-mNeonGreen and pCS107-GFP. pCS2-Plekhg5 was a kind gift from Chenbei Chang, University of Alabama at Birmingham, Birmingham, AL, USA (Popov et al., 2018). The plasmid pSP64T-Dvl2-HA-RFP was linearized with *SalI*; pXT7-EGFP-ADIP, pCS2-mNeonGreen-Shroom3, pCS2-myr-tagBFP-HA, pCS2-myrGFP, and pCS2-Dvl2-GFP were linearized with *Not I*; pCS105-HA-RFP-Diversin, pCS107-GFP-Diversin, and pCS2-Plekhg5 were linearized using *Hpa I*. Capped mRNAs were synthesized from linearized plasmid templates using the mMessage mMachine SP6 Transcription Kit (Invitrogen) and purified with the RNeasy Mini (Qiagen). DivMO (5′-GGC CAC ATC CTG CTG GCT CAT GAA T-3′) (Yasunaga et al., 2011) and CoMO (5′-GCT TCA GCT AGT GAC ACA TGC AT-3′) (Reis et al., 2021) were described previously.

### *Xenopus* embryos, microinjections and wound healing assay

Wild-type *Xenopus laevis* were purchased from Xenopus 1 (MI, USA) and maintained in accordance with the guidelines of the Institutional Animal Care and Use Committee (IACUC) at the Icahn School of Medicine at Mount Sinai. *In vitro* fertilization and embryo culture were carried out as previously described (Ossipova et al., 2014), and embryonic staging was determined according to Nieuwkoop and Faber (1994). RNA or morpholino was injected in 5-10 nl of RNase-free water (Invitrogen) into one or two animal or dorsal blastomeres at the four-cell stage. Wound healing assays for ADIP+Dvl2 co-injected embryos were performed as previously described (Chu et al., 2025). For assays involving the Dvl2 dominant-interfering construct, four-cell stage *Xenopus* embryos were injected with 1 ng of Dvl-DEP+ mRNA into all four blastomeres. At stage 9.5, ectodermal explants were isolated, and wound closure was assessed 1 h post-wounding.

### Protein localization analysis, immunoprecipitation and immunoblotting

To examine Diversin localization, 100 pg of RFP-Diversin mRNA and 50 pg of myr-BFP mRNA were co-injected into a ventral-animal blastomere at the four-cell stage. For experiments investigating polarization triggered by apical constriction, the right ventral-animal blastomere was injected with 100 pg of Diversin mRNA and 50 pg of myr-BFP, while the left ventral-animal blastomere was targeted with 20 pg of mNeonGreen-Shroom3 or *Plekhg5* mRNA together with 20 pg of myrGFP mRNA. For knockdown experiments, one dorsal-animal blastomere was injected with either 30 ng DivMO or 30 ng CoMO, along with 50 pg of HA-tagged myr-BFP RNA. To assess the role of Diversin in planar polarity within the neural ectoderm, 250 pg RFP-Diversin was unilaterally injected into dorsal blastomeres at the four-cell stage. Injected embryos were cultured in 0.1× Marc's Modified Ringer's (MMR) solution at 12-14°C until the desired developmental stages (early gastrula to neurula). Each experimental condition included at least 15-20 embryos, and all experiments were independently replicated a minimum of three times.

Immunoprecipitations and immunoblotting have been carried out as previously described (Itoh et al., 2021). Four-cell embryos were injected into the animal pole of four-eight-cell embryos with RNAs encoding Dvl2-GFP (200 pg), and HA-RFP-ADIP, (400 pg). Embryos were lysed in 1% Triton X100 lysis buffer and the protein was assessed in embryo lysates precipitated with GFP-trap (Chromotech). Immunoblotting was carried out with monoclonal anti-GFP (Santa-Cruz Biotechnology) and anti-HA (12A5) antibodies. Chemiluminescence signals were measured using ChemiDoc (Bio-Rad).

### Immunostaining and imaging

Embryos were fixed in MEMFA for 1 h at room temperature for F-actin visualization using Alexa Fluor 555-conjugated phalloidin (Thermo Fisher Scientific). Fixed embryos were permeabilized with 0.1% Triton X-100 in PBS for 10 min, then incubated overnight at 4°C in phalloidin diluted 1:500 in PBS containing 1% BSA. For immunostaining of HA and Vangl2, embryos were fixed in 2% trichloroacetic acid (TCA) for 30 min at room temperature, permeabilized in 0.3% Triton X-100 for 30 min, and incubated with primary antibodies: mouse anti-HA (12A5) and rabbit anti-Vangl2 (Ossipova et al., 2015b). Alexa Fluor 488-conjugated goat anti-mouse (Thermo Fisher Scientific) and Cy3-conjugated donkey anti-rabbit IgG secondary antibodies (Jackson ImmunoResearch). Fluorescence imaging was performed using a BC43 spinning disk confocal microscope (Andor, Oxford Instruments) with a 20× objective and Fusion Version 2 software. Z-stacks were acquired and processed as maximum intensity projections using ImageJ. All experiments were replicated in at least three independent experiments, each with a minimum of five embryos per condition.

### Quantification and statistical analysis

*Xenopus* ectodermal cell images were segmented using Cellpose (Stringer et al., 2021). Cell apical domain size was quantified by importing segmented cell masks into ImageJ using the imagej_roi_converter.py macro from the Cellpose GitHub repository and individual cell areas were measured. The resulting values were used to generate scatter dot plots in GraphPad Prism version 10.5.0. Cellpose-segmented masks of planar-polarized Diversin were further analyzed for protein orientation and polarity as described in Chu et al. (2025). Briefly, protein aggregates were log-transformed, normalized, and converted into a binary mask using a 20% threshold. Protein clusters were identified using OpenCV (opencv-python 4.10.0.84, Connected components algorithm), and filtered by size exclusion of clusters smaller than two pixels. Unit vectors from each cell center to protein cluster centroids were computed and averaged, weighted by protein cluster size or intensity, to quantify cell polarity. Polarity orientation was calculated as the angle between the resulting polarity vector and a force vector directed toward neighboring constricting cells. Rose plots were generated by computing the angular orientation of protein clusters relative to a reference axis, with angles weighted by cluster size or intensity and binned into circular histograms to visualize planar polarity distributions. Statistical significance was assessed using the Chi-square test.

### Acknowledgements
We thank Jakob Schauser and Ala Trusina for advice with image analysis and quantification, Steph Yang for help with Diversin imaging in the neural plate. We are grateful to members of the Sokol laboratory for valuable discussions.

### Competing interests
The authors declare no competing or financial interests.

### Author contributions
Conceptualization: S.Y.S., S.S.V.; Formal analysis: S.S.V.; Funding acquisition: S.Y.S.; Investigation: S.S.V., C.-W.C., K.I.; Methodology: S.Y.S., S.S.V., C.-W.C., K.I.; Supervision: S.Y.S.; Validation: S.S.V., C.-W.C., K.I.; Visualization: S.S.V., C.-W.C.; Writing – original draft: S.Y.S., S.S.V.; Writing – review & editing: C.-W.C., K.I.

### Funding
This research was supported by the National Institute of General Medical Sciences grant R35GM122492 and National Institute of Child Health and Human Development grant R01HD092990 to S.Y.S. Open Access funding provided by Icahn School of Medicine at Mount Sinai. Deposited in PMC for immediate release.

### Data and resource availability
Raw data and images have been deposited in the Mendeley Data and are publicly available at https://data.mendeley.com/datasets/2mjtgr59kz/1.

### Peer review history
The peer review history is available online at https://journals.biologists.com/bio/lookup/doi/10.1242/bio.062128.reviewer-comments.pdf.

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
