## [Peer Review File · Biology Open]

Mechanosensitive localization of Diversin highlights its function in vertebrate morphogenesis and planar cell polarity

Satheerja Santhi Velayudhan, Chih-Wen Chu, Keiji Itoh and Sergei Sokol

DOI: 10.1242/bio.062128

Editor: Tristan Rodriguez

Review timeline

Original submission: 23 June 2025

Editorial decision: 30 June 2025

First revision received: 6 July 2025

Accepted: 22 July 2025

Original submission

First decision letter

MS ID#: bio.062128

MS Title: Mechanosensitive localization of Diversin highlights its function in vertebrate morphogenesis and planar cell polarity

Authors: Satheerja Santhi Velayudhan, Chih-Wen Chu, Keiji Itoh and Sergei Sokol

I have now reached a decision on the above manuscript.

The reviewer reports are shown at the bottom of this email or can be accessed, together with a copy of this decision letter, by going to:

As you will see, the reviewers gave favourable reports, but raised some minor critical points that will require amendments to your manuscript. I hope that you will be able to carry these out, because we would like to be able to accept your paper.

At this stage, we also ask you to ensure your manuscript complies with our formatting guidelines "please see our manuscript preparation guidelines for details. Provided you are able to fully address the referees' comments, we are positive about publication of your paper (we accept over 95% of revision submissions) and therefore hope you won't mind any extra work involved in reformatting your manuscript at this point.

Please ensure that you clearly highlight all changes made in the revised manuscript. Please avoid using 'Tracked changes' in Word files as these are lost in PDF conversion.

I should be grateful if you would also provide a point-by-point response detailing how you have dealt with the points raised by the reviewers in the 'Response to Reviewers' box. Please attend to all of the reviewers' comments. If you do not agree with any of their criticisms or suggestions please explain clearly why this is so.

Reviewer 1

Comments for the author

This manuscript explores the role of Diversin, a *Xenopus* orthologue of the fly protein Diego, which functions within the Planar Cell Polarity (PCP) signaling pathway—a connection supported by prior research cited by the authors. The study builds on recent findings from the Sokol lab, which revealed an interaction between Diversin and the microtubule-binding protein ADIP. Given ADIP's involvement in mechanotransduction, the authors propose that Diversin may contribute not only to PCP signaling but also to epithelial cells' responses to mechanical forces.

To test this hypothesis, the authors combine loss-of-function with over-expression experiments in *Xenopus* embryos. The morpholinos used for the LOF experiments have previously been characterised and published, which demonstrated their efficiency in depleting Diversin protein. By analyzing the expression of PCP proteins and ADIP—both of which interact with Diversin—they demonstrate that Diversin- and Dvl2-containing PCP complexes regulate morphogenesis in a tension-dependent manner.

The manuscript is exceptionally well-written and features outstanding illustrations, with images of remarkable clarity and quality. Even for non-experts, the narrative is accessible, and the data robustly support the authors' conclusions. Overall, this is an excellent and compelling study.

Minor Comments:

While the manuscript mentions the induction of neural tube defects (NTDs), these are not visually depicted. Including a supplementary figure showing these NTDs could further strengthen the study.

Additionally, some constructs used in the experiments, such as myr-BFP, could benefit from a brief explanation in the Results or Methods section. Clarifying whether the proteins of interest are expressed as fusion proteins with fluorescent tags or contain IRES sequences would also aid reader comprehension.

Regarding Figure 1, the legend states: "(B-C) Diversin depletion inhibits neural plate folding..."—yet it is unclear whether this effect is directly visible in the figure. Furthermore, the difference in myr-BFP expression between panels D and F warrants an explanation in the legend for better clarity.

Reviewer 2

Comments for the author

Velayudhan et al, describe the mechanosensitive role for Diversin in *Xenopus* PCP morphogenesis. The authors demonstrate that Diversin exhibits tension-dependent localisation in during neurulation and during wound healing. It complements this groups recent study on ADIP mechanosensitive nature (PMID: 40562038). The work adds more molecular insight into how mechanical cues interface with PCP.

Experiments are well performed, with proper controls. The DivMO morpholino that the group have used previously can be rescued by over-expression of WT (and not MO-resistant) Diversin and is described in the Yasunaga et al., 2011 paper. Analysis is done well.

N values are provided under plots. I would also suggest that the raw data be made available. The methods are nicely presented.

Overall this is a nice study that is well-supported and well-articulated. One possible inclusion in the discussion is the recent finding in from the Wallingford lab Weng et al., 2025: PMID 40222643). This and the recent paper from the inner ear (Prakash et al, 2025: PMID 40280944) can give another possible way to measure the effects of Diversin knock-down - asking whether it's localisation correlates with polarised junctional fluctuations. It would be useful to incorporate this possibility in the discussion.

The Sokol group have a deep understanding of the role of mechanics in PCP, and so as would be expected, the article provides good background and on the whole discusses the work in the context

of the literature well. Aside from the two exclusions highlighted above, it might be useful to compare the Diversin MO phenotype with the Ankrd6 phenotype from the mouse.

Reviewer's Responses to Questions

Experimental quality

Does each figure have the proper controls?

If 'No', please indicate reasons in Comments for Author box below.

Reviewer #1:

- Yes

Reviewer #2:

- No

Were the data analyzed using appropriate statistical tests?

If 'No', please indicate reasons in Comments for Author box below.

Reviewer #1:

- Yes

Reviewer #2:

- Yes

Reproducibility

Were experiments performed using adequate number of biological replicates?

If 'No', please indicate reasons in Comments for Author box below.

Reviewer #1:

- Yes

Reviewer #2:

- No

Does the methods section provide sufficient detail to permit reproducibility?

If 'No', please indicate reasons in Comments for Author box below.

Reviewer #1:

- Yes

Reviewer #2:

- Yes

Completeness

Are the manuscript's conclusions supported by the data?

If 'No', please indicate reasons in Comments for Author box below.

Reviewer #1:

- Yes

Reviewer #2:

- No

Scholarship

Do the authors cite and discuss the merits of data that would argue for and against their conclusion?

If 'No', please indicate reasons in Comments for Author box below.

Reviewer #1:

- Yes

Reviewer #2:

- No

Does the manuscript title & abstract accurately reflect the contents of the manuscript, without hyperbole?

If 'No', please indicate reasons in Comments for Author box below.

Reviewer #1:

- Yes

Reviewer #2:

- No

First revisionAuthor response to reviewers' comments

Reviewer 1: This manuscript explores the role of Diversin, a *Xenopus* orthologue of the fly protein Diego, which functions within the Planar Cell Polarity (PCP) signaling pathway—a connection supported by prior research cited by the authors. The study builds on recent findings from the Sokol lab, which revealed an interaction between Diversin and the microtubule-binding protein ADIP. Given ADIP's involvement in mechanotransduction, the authors propose that Diversin may contribute not only to PCP signaling but also to epithelial cells' responses to mechanical forces.

To test this hypothesis, the authors combine loss-of-function with over-expression experiments in *Xenopus* embryos. The morpholinos used for the LOF experiments have previously been

characterised and published, which demonstrated their efficiency in depleting Diversin protein. By analyzing the expression of PCP proteins and ADIP—both of which interact with Diversin—they demonstrate that Diversin- and Dvl2-containing PCP complexes regulate morphogenesis in a tension-dependent manner.

The manuscript is exceptionally well-written and features outstanding illustrations, with images of remarkable clarity and quality. Even for non-experts, the narrative is accessible, and the data robustly support the authors' conclusions. Overall, this is an excellent and compelling study.

Minor Comments:

While the manuscript mentions the induction of neural tube defects (NTDs), these are not visually depicted. Including a supplementary figure showing these NTDs could further strengthen the study.

Following this request, we added three new panels to Fig. 1 (D, E, E'), showing unilateral neural tube folding defects in DivMO-injected embryos as compared to a control MO-injected embryo.

Additionally, some constructs used in the experiments, such as myr-BFP, could benefit from a brief explanation in the Results or Methods section. Clarifying whether the proteins of interest are expressed as fusion proteins with fluorescent tags or contain IRES sequences would also aid reader comprehension.

We have now added a brief explanation in the Methods. "pCS2-myr-tagBFP-HA was generated by combining the membrane-targeting signal peptide from the Lyn kinase (MGC1KSKRKDNLNDDE), mTagBFP2 and three HA tags (YPYDVPDYA) in pCS2" (Matsuda et al., 2023)".

Regarding Figure 1, the legend states: "(B-C) Diversin depletion inhibits neural plate folding..."—yet it is unclear whether this effect is directly visible in the figure.

As described above, new Fig. 1E, E' show unilateral neural tube folding defect in DivMO-injected embryo as compared to control MO-injected embryo in Fig. 1D.

Furthermore, the difference in myr-BFP expression between panels D and F warrants an explanation in the legend for better clarity.

Revised Fig. 1 legend states that the "differences in myr-BFP expression reflect variations in the location of the injection site". These differences do not affect our conclusions.

Reviewer 2: Velayudhan et al, describe the mechanosensitive role for Diversin in *Xenopus* PCP morphogenesis. The authors demonstrate that Diversin exhibits tension-dependent localisation in during neurulation and during wound healing. It complements this groups recent study on ADIP mechanosensitive nature (PMID: 40562038). The work adds more molecular insight into how mechanical cues interface with PCP.

Experiments are well performed, with proper controls. The DivMO morpholino that the group have used previously can be rescued by over-expression of WT (and not MO-resistant) Diversin and is described in the Yasunaga et al., 2011 paper. Analysis is done well.

N values are provided under plots. I would also suggest that the raw data be made available.

Prompted by the reviewer, raw data associated with Fig. 1, 2, 5, have been deposited to Mendeley Data, and are publicly available at [https://doi: 10.17632/2mjtgr59kz.1](https://doi.org/10.17632/2mjtgr59kz.1).

The methods are nicely presented.

Overall this is a nice study that is well-supported and well-articulated. One possible inclusion in the discussion is the recent finding in from the Wallingford lab Weng et al., 2025: PMID 40222643). This and the recent paper from the inner ear (Prakash et al, 2025: PMID 40280944) can give another possible way to measure the effects of Diversin knock-down - asking whether it's localisation

correlates with polarised junctional fluctuations. It would be useful to incorporate this possibility in the discussion.

We revised the discussion to incorporate the recent publications suggested by the reviewer. The text states that “Together with the recent reports, in which PCP correlates with the anteroposterior differences in junctional tension in frog mesoderm (Weng et al., 2025) and myosin light chain phosphorylation in the chick inner ear (Prakash et al, 2025), our work further strengthens the link between PCP and tensile forces in early development.”

The Sokol group have a deep understanding of the role of mechanics in PCP, and so as would be expected, the article provides good background and on the whole discusses the work in the context of the literature well. Aside from the two exclusions highlighted above, it might be useful to compare the Diversin MO phenotype with the Ankrd6 phenotype from the mouse.

To draw a parallel between our results and the reported genetic phenotypes in the mouse, we added the following sentence to the first paragraph of the discussion. “Supporting a conserved role for Diversin/Ankrd6 in PCP, mice lacking the *ankrd6* gene exhibited PCP defects in the hair cells of the inner ear (Jones et al., 2014)”.

Second decision letter

MS ID#: bio.062128R1

MS Title: Mechanosensitive localization of Diversin highlights its function in vertebrate morphogenesis and planar cell polarity

Authors: Satheeja Santhi Velayudhan, Chih-Wen Chu, Keiji Itoh and Sergei Sokol

I am happy to tell you that your manuscript has been accepted for publication in Biology Open, pending our standard publication integrity checks. It was accepted on 22 July 2025.